# Measuring the Effectiveness of Career Education: A Kindergarten Intervention

Alessandro Buffoli [1], Teresa Rinaldi [2], Roberta Morici [3] and Diego Boerchi [4,*]

1 Faculty of Education, Università Cattolica del Sacro Cuore, 20123 Milan, Italy
2 Faculty of Psychology, eCampus University, 22060 Novedrate (CO), Italy
3 Faculty of Psychology, Università Cattolica del Sacro Cuore, 20123 Milan, Italy
4 Department of Psychology, Faculty of Education, Università Cattolica del Sacro Cuore, 20123 Milan, Italy
* Correspondence: diego.boerchi@unicatt.it

**Abstract:** Several studies have confirmed the importance of career education in promoting career development in children. This study aims to test whether specific career education interventions would develop new conceptions about career choice and career attainment in kindergarten pupils. The intervention was conducted directly by teachers who were adequately trained and supervised. The career conceptions were assessed in experimental and control groups through the Conceptions of Career Choice and Attainment protocol, before and after career education intervention. The results showed that the two groups started from the same level and increased their conceptions over time. However, the experimental groups increased them much more, and statistically significantly, than the control groups.

**Keywords:** career development; career conceptions; career education; kindergarten pupils; educational intervention; measurement

## 1. Introduction

The study presented in this article aims to test the validity of an intervention related to career choice development specifically created for kindergarten pupils in northern Italy. This work follows the research thread that aims to build preventive, effective, and exportable instruments to improve career choices. The specific relevance of this study is a result of the lack of existing interventions for such an early age group that can create representation and refine reasoning in regard to career education. In addition, the intervention includes training and the subsequent implementation of the intervention by classroom teachers to foster the development of good practices to be applied in daily work. It also aims to support anyone involved in career guidance and training to build theoretically solid and easy-to-implement interventions.

First, a theoretical introduction will be presented concerning the main intervention models used in the literature that promote career choice.

Next, a description of the Conceptions of Career Choice and Attainment model, the theoretical framework for this specific intervention, and an overview of the types of interventions regarding career education used in school age will be given.

Then, the research will be described in detail, including the methodology, results, and discussion.

### 1.1. Theoretical Background

Researchers assume that social, cultural, and historical contexts are important in the career development process and impact people's career choices. The literature offers new models of theory integration and suggested areas [1–4]. In the career development literature, the Social Cognitive Career Theory tried to explain how people develop interests, make and implement career choices, and achieve satisfactory study and work performance [5–8].

Other theories on vocational development have focused on how career choices change with age [9,10]. They recognized that career development is a lifelong process rooted in childhood [11–13], adopting a lifespan perspective [14,15] and identifying the evolutionary stages and tasks [16,17].

Different theories generally consider career development as a continual process of change and adaptation that begins in childhood [14,16,18,19] and is influenced by individual and contextual factors. They affect the aspirations and expectations that are developed in relation to studies and the world of work [19]. Furthermore, several studies have confirmed the role of career education in promoting career development in children [20–24].

The main objective of this study is to test whether a specific career education intervention, designed for the particular target of kindergarten pupils, helps foster the development of more mature career choices conceptions, according to the Conceptions of Career Choice and Attainment model [25].

## 1.2. Children's Career Development Theories

Particular attention has been directed at the effectiveness of career development interventions during the education path. By integrating the psychological knowledge perspective and the social conditions change analysis, increasingly updated models of understanding and intervention have been theorized [26]. For example, studies have shown that, in this developmental period, children increasingly develop skills of self-awareness [27,28], defining the representations they build of themselves, the world and their relationships to it [29,30]. With the arrival of hypothetical thinking [31,32], children engage in increasingly conscious reasoning about the future and what they want to become [33]. Through representations, children increasingly reason about different job types, what is needed to be accepted for that job and the benefits and risks of having a specific occupation [1], using that information for their future career paths [34]. Even if the contributions to the child population are minimal [2], the skills they acquire, if supported by theoretical models and structured interventions, promote self-management and career development-related skills and development [35]. The most recent intervention models that have received the most interest among researchers are the constructivist "Life Design" of Savickas and co-workers [19], the socio-cognitive theory of Lent, Brown and Hackett [6–8] and the evolutionary model of career choice and achievement [25].

## 1.3. Howard and Walsh's Conceptions of Career Choice and Attainment (CCCA) Model

Reviews of the literature on career development [35,36] highlighted the major attention given to children's vocational expectations and aspirations, as well as their knowledge of the world of work [25]. Less studied is how children think and understand the work world [37]. Career development research has highlighted the importance of beginning career work with children before adolescence [36,38,39]. Studies have shown that children as young as four years old reflect on the suitability of careers [39]. On the other hand, few career theories address the children's needs, and those that do are fragmented [40]. According to Howard and Walsh [25], starting from cognitive-developmental psychology, it is possible to explain the maturation of the processes that lead to choosing and obtaining a future job; in fact, this approach allows one to understand the way people build their understanding of the world [40]. This approach has been used to understand children's conceptions of different complex phenomena [41–43], and it has been shown that such conceptions would be formed following a developmental sequence [25].

Starting from these premises, Howard and Walsh [25] proposed the Conceptions of Career Choice and Attainment (CCCA) model to understand children's conceptions of career processes. They underlined how children and young people conceptualize key career development processes, focusing in particular on two fundamental processes of career development: the choice and the attainment of a career/job. Three progressively more refined approaches are identified in the CCCA, used throughout development to understand the career choice and achievement processes. These processes are, in turn,

divided into two levels, used by children to reflect on the work world and that explain the development of vocational thinking [25,44].

The first approach is Association [25]: children focus on specific activities, experiences, and directly observable objects related to a particular profession (for example, the worker's uniform or workplace). Moreover, children who adopt this approach often base the choice of their future work on imagination, fantasy, and heroism, without a fundamental comprehension of the processes [25]. Within the Association approach, we can distinguish the following two levels. In Level 1, Pure Association: for children, work simply exists, and they are unable to explain how individuals select career options, people simply go to work. At this level, career concepts consist of a confusing list of known notions about a particular profession [25]. Level 2, Magic Thinking, is characterized by the importance of appearances: children believe that to obtain a job, it is enough to possess ordinary objects belonging to that job (e.g., the doctor's case for becoming a clinician), with any consideration of the skills needed for the job. Children at this level also assert the existence of a simple method to make choices and attain a specific career. However, they do not explain a mechanism for implementing this method or the skills and features required to exercise a particular profession [25].

The second approach is Sequence, and it emphasizes the mechanism or order through which people choose and obtain a particular job [25]. According to this approach, children conceptualize the two processes of career choice and attainment as separate, and they can explain in concrete terms how these processes are related. Certain aspects of self, such as interests and gender, are considered and compared with known work activities. Children provide arguments based on people's involvement in activities that lead to a specific job. For example, they may think that to become a chemist, people first need a high school degree, then a university degree, and then some work experience in a lab. In fact, in this approach, children also argue that there is a spatial contact or linear time sequence that inevitably leads to work: for example, children believe that once people graduate and go to university, they can start working [25,44]. In the Sequence approach, there are also two levels. In Level 3, External Activities, children describe a simple process by which people acquire knowledge about professions and choose the one they prefer: job choice depends on what one likes, generally identified by the more macroscopic aspects of individual professions. Automaticity characterizes this level: children assume that, by taking the necessary steps, people will automatically obtain the job according to the observable skills and activities required to achieve it [25]. However, at this level, children still do not have a clear idea about how the transition from one level of education to another can lead to a job: children believe that it is enough to follow the educational process to deterministically achieve the desired career [25]. In Level 4, Internal Processes and Capacities, children begin to match themselves to a job. They describe the career choice process as a correspondence between themselves and a job: in addition to the activities and characteristics of the job, personal interests and skills are considered. Moreover, the achievement of the job depends on learning specific skills and possessing the physical abilities required to perform that job [25].

The third approach is Interaction: adolescents who use this approach believe that the processes of career choice and achievement are determined by the complex and dynamic interactions between the individual's characteristics (interests, skills, predispositions, values) and the environmental influence (opportunities to develop particular skills, job availability, labor market conditions). This approach assumes the typically adolescent acquisition of formal thinking. The latter allows individuals to think about abstract constructs, such as their professional values, and achieve a more precise and articulated understanding of the different reputations the professions may enjoy. Moreover, they fully understand that similar processes may conduct different outcomes [25]. In the Interaction approach, there are also two levels. In Level 5, Interaction, adolescents reflect on the processes of choice and achievement of a career and identify a dynamic interaction between individual (biological and psychological), relational (referring to interaction with others) and environmental

(characteristics and availability of work) causes. They consider immediate environmental causes, such as job skills and job availability. In Level 6, Systemic Interaction, the causes mentioned above are integrated with systemic factors, such as employment trends, labor market conditions, and the appearance of new areas of employment. It happens because the more abstract and general labor market elements are also incorporated. The interaction between aspects of self and micro- and macro-environmental inputs becomes dynamic.

### 1.4. Career Education Interventions in Schools

The literature includes few experimental career education interventions in schools. However, several studies conducted in secondary schools have demonstrated the effectiveness of career education interventions [45–48]. There are also studies implemented in primary schools [49] and dissertations on the importance of approaching career development from the early school years [50].

Within the Italian context, Ginevra and Nota [51] proposed a program called "Journey in the world of professions and work" for primary school students. This program was delivered by a counsellor, unaware of the purposes of the study and the formulated assumptions, to 154 children in two schools in a province in northern Italy. The program included ten teaching units, each lasting 2 h a week for ten weeks. The aim was to promote children's concern for a career characterized by hope and optimism towards the future, curiosity [52], career exploration, and a knowledge of professions. For each unit, the objective has been described, specific activities have been planned, and the administration of ten multiple-choice questions related to the topic have been considered. As suggested by Nota and colleagues, [53] and Savickas in 2009 [54], high relevance has been given to the active participation of children—through the use of group work, stories and exercises—and to the use of narration as a tool to reflect and describe personal situations [37,55]. The study's results revealed an improvement in optimism, hope, curiosity, and concern for careers, as well as an increase in knowledge about professions and planning skills in the experimental group.

## 2. Materials and Methods

### 2.1. Aim

This study aims to test whether an intervention designed for kindergarten pupils can foster the development of more well-articulated career-related thinking. The CCCA model [49] was used to accomplish this.

### 2.2. Sample

The participants included 58 children attending three kindergartens in a medium city in the north of Italy. The activities were designed for class participation, so assigning the children randomly to the experimental and the control groups was impossible. Two of the four classes involved were randomly defined as experimental, and the others as a control group. They were all enrolled in this study at the beginning of their 2nd grade (4 years old) and followed until the end of their 3rd grade (5–6 years old). The two groups were not equal by gender distribution: females were more represented in the experimental group (72%) and reasonably balanced in the control group (41%). Parental informed consent was obtained as well as verbal assent from the children, and the interviewer recorded their responses. The research was conducted according to the APA ethical standards.

### 2.3. Study Design

This study is part of a research project between the guidance services of a municipality in northern Italy and the Center for Research on Vocational Guidance and Socio-professional Development (CROSS) of the Università Cattolica del Sacro Cuore in Milan. The experimenters, teachers and psychologists from the guidance services, previously trained in the CCCA model, designed the following two-year intervention. The study was organized into three steps:

1. Pretest: every child responded verbally to the Conceptions of Career Choice and Attainment (CCCA) Protocol;
2. Intervention: the experimental group intervention consisted of meeting workers with different jobs, family members or unknown participants. Most importantly, to stimulate concern and curiosity, the teachers helped their pupils to prepare interviews with the workers they were about to meet. The students were also asked to investigate the reasons they chose that job and what they did to train and achieve it. Considering the children's development phase, the purpose was to integrate the motives that they would have considered on their own with those of other people, not to discuss the correctness of the reasons they have listened to, letting them experience that the motivations can be very different, and should arise from inner needs and reflections that may be different in every person. The interviews were conducted in economic activities or kindergarten. Finally, reflections and consolidation of the experience were allowed by a session of circle time and the creation of play corners themed according to the jobs considered. For both the control and experimental groups, the teachers planned visits to local manufacturing and business sites to allow the children to explore professions, starting from the context in which they are placed. In the following days, only the children in the experimental group, using materials provided by the teachers and different techniques, made graphic elaborates related to the significant elements of the trip, training their creativity. Afterwards, the teachers asked them to describe the jobs and, where possible, the verbalizations were annotated to monitor the development of expressive ability, communication skills, observation, and reflection skills. At the end of the first year of the intervention, the CCCA Protocol was repeated with every child as a mid-term review;
3. Posttest: at the end of the second year of intervention, corresponding to the end of the kindergarten cycle, the CCCA Protocol was repeated with every child.

Every time the interviewer ran the CCCA Protocol, the children were in a quiet room different from the children's classes. The intervention was conducted in classrooms.

### 2.4. Measure the Conceptions of Career Choice and Attainment (CCCA) Protocol

The protocol used for the interviews was requested by Howard and colleagues, as explicated in the Supplementary materials associated with their 2010 article [48]. Generally, it consists of a semi-structured interview, the duration of which varies depending on the awareness level of the children's responses. The seven open-ended questions follow the two poles of the model: career choice and job attainment. Specifically, they cover the types of jobs known, what their preference is, and the path to reach that job position. The researchers obtained the Italian version of the protocol through back translation. To test the adequacy of the translation, a small group of kindergarten pupils not involved in the project were asked to answer the questions.

## 3. Results

As the collected data were ordinal, we used the Mann–Whitney U statistic to separately test the potential differences among assessments [56]. Group belonging served as the independent variable, while the frequencies observed for each level designation on the CCCA served as the dependent variable.

### 3.1. Conception of Career Choice

The two groups did not differ before the intervention as the number of students assigned to levels 1 and 2. The children in level 1 typically responded "I don't know" to most questions. An example of the children's answers belonging to level 2 (representing Magical Thinking) was: "I want to be a police officer when I grow up./You must have the hat, the one to give tickets, the little sheet" or "He has to wear shoes and a blue suit and have a police car." The number of children assigned to levels 2 and 3 increased after the

first year and, slightly further, at the end of the second year. This was more consistent in the experimental group (Table 1).

**Table 1.** Number and percentages of students for conceptions of career choice by group.

| | Level 1 | Level 2 | Level 3 |
|---|---|---|---|
| Pre-intervention (U (1, 58) = 391.500, z = −0.523, *p* = 0.601) | | | |
| Experimental group | 15 (51.7%) | 14 (48.3%) | 0 (0.0%) |
| Control group | 17 (58.6%) | 12 (41.4%) | 0 (0.0%) |
| End of the first year (U (1, 58) = 285.500, z = −2.589, *p* = 0.010) | | | |
| Experimental group | 2 (6.9%) | 25 (86.2%) | 2 (6.9%) |
| Control group | 12 (41.4%) | 15 (51.7%) | 2 (6.9%) |
| End of the second year (U (1, 58) = 281.500, z = −2.818, *p* = 0.005) | | | |
| Experimental group | 1 (3.4%) | 22 (75.9%) | 6 (20.7%) |
| Control group | 7 (24.1%) | 21 (72.4%) | 1 (3.4%) |

*3.2. Conception of Career Attainment*

In addition, in this case, the two groups did not differ before the intervention as the number of students assigned to levels 1 and 2. After the first year, the number of children assigned to levels 2 and 3 increased consistently for the experimental group and slightly for the control one. An example of children assigned to level 3, in which reflections about career choice begin to be sequential, was: "I want to be a hairdresser. There is a course where you learn how to cut hair well, do color, and wash hair well." At the end of the second year, the differences between the two groups were less consistent because the experimental group increased less than the control (Table 2).

**Table 2.** Number and percentages of students for conceptions of career attainment by group.

| | Level 1 | Level 2 | Level 3 |
|---|---|---|---|
| Pre-intervention (U (1, 58) = 449.500, z = 0.548, *p* = 0.584) | | | |
| Experimental group | 20 (69.0%) | 9 (31.0%) | 0 (0.0%) |
| Control group | 18 (62.1%) | 11 (37.9%) | 0 (0.0%) |
| End of the first year (U (1, 58) = 266.500, z = −3.011, *p* = 0.003) | | | |
| Experimental group | 1 (3.4%) | 27 (93.1%) | 1 (3.4%) |
| Control group | 13 (44.8%) | 14 (48.3%) | 2 (6.9%) |
| End of the second year (U (1, 58) = 292.500, z = −2.660, *p* = 0.008) | | | |
| Experimental group | 1 (3.4%) | 21 (72.4%) | 7 (24.1%) |
| Control group | 5 (17.2%) | 23 (79.3%) | 1 (3.4%) |

**4. Discussion**

Starting from the results, the experimental and control group did not differ before the intervention. The number of students assigned to Levels 1 and 2, according to Howard and Walsh's CCCA model and protocol [46], was similar. This means that, procedurally, the two groups were balanced, while developmentally, they were comparable. This result is in line with the classical theories of psychological development that suggest how the reflective level of children aged 3 to 4 years is solely related to their concrete experience of the world.

It was unnecessary to wait until the end of the intervention, as already by the end of the first year, the children in the experimental group showed greater maturity in thinking about the inherent career thinking. In fact, the number of children in levels 2 and 3 displaying career choice conceptions increased after the first year and slightly further at the end of the second year, more consistently so for the experimental group. In addition, regarding career attainment conceptions, the number of children assigned to Levels 2 and 3 after the first year increased consistently for the experimental group and just slightly for the control one. At the end of the second year, the differences between the two groups were still present, but a little less consistent. Considering that the control group participated in activities similar to those proposed to the experimental group, the results suggest that only

a structured intervention, such as the one dedicated to the experimental group, can foster career development consistently concerning a mere exposure of children to work contexts.

The validity of the results is also determined by the fact that the pre- and post-test phases were conducted by an external researcher who was unaware of the children's group assignment. In addition, the CCCA protocol, is, by construction, highly codified, which results in a considerable and significant reduction in the influence of the experimenter.

## 5. Conclusions

This study helps to confirm the CCCA model's validity and application in the Italian context. At the beginning of the study, the kindergarten pupils were attributed to levels consistent with their age, according to a previous study by Howard e Walsh [45]. Further validation came from the natural development of the children belonging to the control groups from the beginning to the end of the interventions, a period of nine months, confirming our expectations, which were also supported by the previous literature concerning children's career development [16,17].

A limitation to such a study might be that the development and change were not due to the intervention, but to something external that was not intercepted and measured; however, the school being the same for the two groups, having included the control group, and considering that the activities that children participate in outside the school setting are the same for participants in the two groups, there is no reason to think that the change is due to something outside the intervention.

Considering the results, it is necessary to highlight the effectiveness of the teachers' training. As teachers become aware of the psychological and developmental aspects involved in the career choice process, the intervention design becomes more transparent and effective. Generally, whenever scientifically validated models are involved in an intervention design, the desired outcomes are more likely to be achieved.

The differences observed between the experimental and control groups at the end of the protocols confirm that career education interventions, if projected on the solid base of valid psychological models, such as CCCA and SCCT, can contribute to accelerating the development of career management skills starting from very young ages [1,25].

This study's findings can help all of those involved in career guidance understand that designing actions early in the education process fosters the development of more conscious and self-aware children and will prepare them for the future career choices they will have to make.

This study intends to be an attempt at reflection for all those involved in career guidance, particularly in such a developmentally sensitive age group. It is well known in the literature that children's formed beliefs about themselves and the world start from a concrete and pragmatic reflection of the reality around them. This may lead children to decide, through a lack of experience, to orient themselves only toward known, familiar, or mentally represented jobs. Interventions such as this one, on the other hand, encourage broader reflection and experience by engaging the child in a mental activity different from his/her usual one. Teachers and trainers, through the knowledge of such a model, can accompany the child in the creation of more amplified representations of the world and, above all, can be promoters of experiences that would otherwise slow down the process of reworking information that a child faces in development.

A further reflection that enhances the relevance of interventions such as the one presented in this article concerns the fact that career education is an issue of pragmatic and moral concern. For example, in a country such as Italy, in which the NEET phenomenon is particularly relevant compared to other countries in Europe, interventions aimed at choice enable prevention. They provide an alternative to the approaches generally applied to this kind of phenomenon in which there is a tendency to seek solutions when the problems are already manifest.

This study presents some limitation. First, it might be the case that the development and change were not due to the intervention but to something external that was not

intercepted and measured; however, the school being the same for the two groups, having included the control group, and considering that the activities the children participate in outside the school setting are the same for the participants in the two groups, there is no reason to think that the change is due to something outside the intervention.

Second, although the teachers were adequately trained in the theoretical content, it is possible that in creating the interviews together with the children they influenced, from a methodological point of view (perhaps unskilled in conducting such an activity), the students' participation. For this reason, to address this possible bias, it was decided to divide the conduct of the protocol from the intervention between the teachers and experimenters.

**Author Contributions:** Conceptualization, A.B., T.R., R.M. and D.B.; Investigation, A.B.; writing—original draft preparation, A.B., T.R. and D.B.; writing— review and editing A.B., R.M., T.R. and D.B.; supervision, D.B. All authors have read and agreed to the published version of the manuscript.

**Funding:** This research received no external funding.

**Institutional Review Board Statement:** The study was conducted in accordance with the Declaration of Helsinki for studies involving humans.

**Informed Consent Statement:** Informed consent was obtained from the parents of all subjects involved in the study.

**Data Availability Statement:** All data is contained within the article.

**Acknowledgments:** The authors sincerely thank municipality, kindergarten teachers, families, and especially children who participated in this study.

**Conflicts of Interest:** The authors declare no conflict of interest.

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
