# Peer review of "Measuring the Effectiveness of Career Education: A Kindergarten Intervention"

_psych, doi:10.3390/psych5010011_

Round 1

Reviewer 1 Report

- unify the punctuation presenting sources and follow the citation norm;

- use a proper pedagogical terminology - a kindergarten pupil;

Author Response

Dear Reviewer,

thank you for your suggestions that make us improve and make clearer our paper.

As for the topic of model validity first of all, we have provided linguistic and pedagogical uniformity for the age group considered for this work.

The bibliography was reviewed, edited and adjusted in accordance with the proposed journal standards.

Thanks again for the suggestions that we hope will make the work more readable.

Reviewer 2 Report

Thank you for the opportunity to review this paper. The paper has its merits but I would like to highlight the following fundamental concerns of this paper:

1 - The CCCA model was adopted and the authors claimed that this model is suitable to analyze the career choice of kindergarten children. Please provide precise justifications as studies in developmental psychology have shown that the term "children" can refer to a broad age range of younger human beings and at each stage of development can feature great differences in their cognition and specifically their ability to understand abstract concepts like "careeer development". And this sense, please justify and provide with data your research conclusion that the intervention reported in the study is valid. Please clarify whether there are possibilities of misinterpretation of expected results (e.g., placebo effects).

2 - Validity

Research in early childhood studies use specific procedures to extract data due to the fact that kindergarten children are too young to understand complex matters and express themselves clearly and accurately. These procedures are not adopted in the study. For example, how can it be ensured that the children really said what they meant? Was it because there were environmental issues that promoted them to produce a certain result? Were there teacher biases? That is simply because a child likes a particular teacher so the child will respond positively to whatever the teacher says. 

The procedures also involved asking students why they chose the job they wanted. This can be problematic because there was no clear guidance or a decision rule on the basis of why a job is chosen or not. For example, a common problem in youth career guidance is that young people will not choose to pursue a career simply because they dont like it. Here in this example, is where ethical issues emerge. This is because from an educational point of view, shall students be taught that they should just simply choose whatever they want to do and not the opposite? Please clarify and describe clearly how students, specifically kindergarten children can be guided and counseled into such a deep philosophical matter.

Last, career adaptability and adaptability are both used in the paper. Please note the two concepts are different. Career adaptability is not adaptability. This is a common conceptual conflation/misinterpretation. Please clarify.

Also, since this study is about career education, it is useful to make reference to research in educational psychology, which has a thick collection of literature clearly shown that constructivist learning approaches are ineffective. Please justify why it is claimed that career construction and Life Design, which is a constructivist approach in essence should be used. 

Recent studies have also shown Career adaptability is a flawed concept and contains theoretical issues concerning its conceptualization and psychometric properties, please illustrate how these shortcomings are addressed to strengthen your arguments.

Author Response

Dear Reviewer,

thank you for your suggestions that make us improve and make clearer our paper.

As for the topic of model validity first of all, we have provided linguistic and pedagogical uniformity for the age group considered for this work.

Motivations and limitations concerning data interpretation that can justify the results obtained have also been included within the text in the discussion section.

The section concerning bibliographical references has also been standardized by adjusting it to the proposed format.

In relation to the validity of the study, the fact that the data collection in the pre and post phases was conducted by a researcher outside the intervention who did not know the children's membership in the experimental and control groups meant that there was no influence whatsoever in the detection of responses. This reflection was included within the discussion of the results.

Additional reflections were added on how teachers can guide children in orientation.

As for career adaptability, the construct appeared twice within the text, once within the explanation of the Life Design model.

Following his comment, we decided to remove the part of the text that was originally included to make a brief mention of the theoretical background in which these types of interventions usually fit since the journal was not specialized in the field.

Thanks again for the suggestions that we hope will make the work more readable, understandable and suitable for the publication.

Reviewer 3 Report

The quality of manuscript adhere to standard established for the journal

Author Response

Dear Reviewer,

thank you for reading our paper and considering it suitable for publication standards.

Reviewer 4 Report

The article deals with an interesting and relevant topic in contemporary society. Although the authors have not mentioned it, the topic of the article, i.e., career education, is a topic of pragmatic and moral concern which directly refers to the phenomenon of NEET. Given the context of the study (Italy) and the percentage of NEET in the Italian context, I would encourage the authors to add indications on how this intervention could support Italy to deal with this dramatic condition. 

Nevertheless, I have no concerns about the topic of the paper, its originality, and its contents. Let me just mention that one of the points of strength of the manuscript is the testing of an intervention. My main concern regards a) the level of English and linguistic problems, b) the structure of the paper and presentation of the contents, and c) redundancy along the manuscript.

a) With respect to the English level, I strongly encouraged the authors to have an English Native speaker who can proofread their paper. There are not many grammar errors but most of the sentences appear to be difficult to understand. Also, the authors would benefit from proofreading to render the style of the paper more academic and less redundant (see point c).

b) The structure of the paper is far to be clear. Both the abstract and the introduction are vague (with also non-academic ways of saying e.g., "some studies etc..; Many researchers ... etc.). Anyway, the introduction doesn't introduce the contents of the paper. I strongly encourage the authors to revise the introduction by a) stating the context and the problem of your paper in the first paragraph; b) stating the general literature that informed your rationale for research; c) stating the aim of the study while explaining why doing this research is relevant; d) adding an overview paragraph where you introduce the contents of the paper -- since your paper has a very long introduction, having an overview of the contents can help the readers. 

To continue, the literature review takes up a lot of space within the paper. This is okay but why do you report so many references from the literature and introduce so many theories if you don't recall them in the discussion? I mean, once you introduce certain theoretical background you must then discuss how your results contribute to the theory. This is one of the main points of weakness of your paper. Please, address it by reviewing the discussion and explaining how your study confirms or not, contributes or not, to the literature that you mentioned. 

In the results, you repeat the aim of the study. This is meaningless. Rather, you could explain how to pursue your aim.

And this last aspect reminds me to point c) redundancy. There are a lot of repetitions within your paper. For example, you stated the aim/objective of your study four times. The best you can do is to introduce the aim of your study once in the introduction and once in the discussion.

Author Response

Dear Reviewer,

thank you for your suggestions that make us improve and make clearer our paper.

With regard to the issue of NEETs, we were very pleased to consider your reflection and added in the discussion part a commentary on why this type of intervention is effective.

To make it easier to read the article, we have taken your suggestion and included a summary paragraph that can help the reader discover the contents. Also, to make the reading more fluent, some parts of the text have been removed or lightened.

The repetition of the study objective has also been shortened and modified for ease of reading.

An overview of general English has been made to try to be less redundant.

Thanks again for the suggestions that we hope will make the work more readable, understandable and suitable for the publication.

Round 2

Reviewer 2 Report

Reviewer's comments on the previous drafts were only partially answered. 

Author Response

Dear reviewer,

we thank you for your feedback.

Best regards

Reviewer 4 Report

Great job with the article. It is improved from my last revision. Congratulations:)

Author Response

Dear reviewer,

Thank you for your contribution to improving the quality of the paper.